# Design and Analysis of a Deep Learning Ensemble Framework Model for the Detection of COVID-19 and Pneumonia Using Large-Scale CT Scan and X-ray Image Datasets

**DOI:** 10.3390/bioengineering10030363

**Published:** 2023-03-16

**Authors:** Xingsi Xue, Seelammal Chinnaperumal, Ghaida Muttashar Abdulsahib, Rajasekhar Reddy Manyam, Raja Marappan, Sekar Kidambi Raju, Osamah Ibrahim Khalaf

**Affiliations:** 1Fujian Provincial Key Laboratory of Big Data Mining and Applications, Fujian University of Technology, Fuzhou 350011, China; 2Department of Computer Science and Engineering, Solamalai College of Engineering, Madurai 625020, Tamil Nadu, India; 3Department of Computer Engineering, University of Technology, Baghdad 10066, Iraq; 4Amrita School of Computing, Amrita Vishwa Vidyapeetham, Amaravati Campus, Mangalagiri 522503, Andhra Pradesh, India; 5School of Computing, SASTRA Deemed University, Thanjavur 613401, Tamil Nadu, India; 6Department of Solar, Al-Nahrain Renewable Energy Research Center, Al-Nahrain University, Baghdad 64040, Iraq

**Keywords:** CT scan images, chest X-ray, transfer learning, COVID-19 detection, deep learning

## Abstract

Recently, various methods have been developed to identify COVID-19 cases, such as PCR testing and non-contact procedures such as chest X-rays and computed tomography (CT) scans. Deep learning (DL) and artificial intelligence (AI) are critical tools for early and accurate detection of COVID-19. This research explores the different DL techniques for identifying COVID-19 and pneumonia on medical CT and radiography images using ResNet152, VGG16, ResNet50, and DenseNet121. The ResNet framework uses CT scan images with accuracy and precision. This research automates optimum model architecture and training parameters. Transfer learning approaches are also employed to solve content gaps and shorten training duration. An upgraded VGG16 deep transfer learning architecture is applied to perform multi-class classification for X-ray imaging tasks. Enhanced VGG16 has been proven to recognize three types of radiographic images with 99% accuracy, typical for COVID-19 and pneumonia. The validity and performance metrics of the proposed model were validated using publicly available X-ray and CT scan data sets. The suggested model outperforms competing approaches in diagnosing COVID-19 and pneumonia. The primary outcomes of this research result in an average F-score (95%, 97%). In the event of healthy viral infections, this research is more efficient than existing methodologies for coronavirus detection. The created model is appropriate for recognition and classification pre-training. The suggested model outperforms traditional strategies for multi-class categorization of various illnesses.

## 1. Introduction

CT scans, medical imaging, or chest X-rays have been proposed as a viable method for detecting COVID-19 quickly and early. CT scan has demonstrated high sensitivity in identifying COVID-19 at the initial assessment of patients. In severe circumstances, it may efficiently correct RT-PCR false negatives [1,2,3,4]. Nevertheless, this is a complex and time-consuming process, because a professional must interpret the X-rays and CT imaging to establish if an individual is COVID-19 positive. The first diagnostic tool to be used to identify COVID-19 pathology is a chest X-ray. A single chest X-ray image is insufficient for reliable prediction and treatment of COVID-19 [5,6,7,8]. To address this limitation, multiple medical data findings are combined, and predictive classifications are generated, leading to improved accuracy compared to using only a single test image. However, extracting relevant features from COVID-19 images is challenging due to daily fluctuations in characteristics and differences between cases [9,10,11,12]. Many conventional machine learning (ML) techniques have already been applied to automatically classify digitized chest data. Using an SVM classification model, frequent patterns were generated from the pulmonary surface to distinguish between malignant and benign lung nodules. Backpropagation networks have been used to categorize imagery as normal or malignant using a gray-level co-occurrence matrix technique [13,14,15].

Convolutional neural networks (CNNs) can remove valuable features in picture categorization tasks. This feature extraction is performed with transfer learning. Pre-trained approaches collect the general properties of large-scale data such as ImageNet and then apply them to the task. DL algorithms are better than traditional NN models if enough labeled images are available. The CNN model is one of the most common DL algorithms for diagnostic imaging, with excellent results. Despite ML algorithms, the effectiveness of CNNs depends on acquiring feature representation from property imagery. As a result, the obtainability of pre-trained models such as DenseNet, ResNet, and VGG-16 is extremely helpful in this procedure and appears to be extremely interesting for COVID-19 identification using chest CT images and X-ray images. Transferring acquired knowledge from a pre-trained network that has completed one function to a new task is a systematic approach for training CNN architecture. This technique is faster and more efficient because it does not require a large annotation training dataset; as a result, most academics, particularly in the medical field, prefer it.

Transfer learning can be performed in three ways [16,17,18,19,20] as follows: shallow tuning adjusts individuals from the previous classification layer to novel problems while leaving the limitations of the remaining levels untrained; the deep tune function is used to retrain the variables that were before the network from end to end; fine tuning intends to prepare many layers after layer and gradually adjust to learning variables until a critical performance is obtained. Knowledge transfer in detecting X-ray images through an impressively executed fine-tuning procedure.

The motivational aspect of the proposed model, along with the significant contributions, are as follows. The suggested system uses transfer learning to train a model faster. The training models compute the appropriate weights. Tuning in to the work at hand, it is identifying COVID-19. The techniques are stacked to anticipate output class. The meta-model in this system is an individual neuron that correctly forecasts the input category using the outputs. To train a model faster, the proposed system employs a learning approach. The weights of the pre-trained method are correctly adjusted during training, and the approaches are integrated via stacking. The best model architecture and training parameters for COVID-19 and pneumonia diagnosis should be automated in order to produce better results. Using transfer learning methodologies solves research gaps and reduces training time. 

The research is organized as follows: Section 2 focuses on the related existing works for COVID-19 detection cases; Section 3 discusses the proposed work and the details of the algorithms presented in this study; Section 4 illustrates the graphical plots and tabular representation for the experimentation of the proposed and existing methods; Section 5 summarizes the proposed work and concludes with its enhancements.

## 2. Literature Survey and Related Work

The unique stacked ensemble model is proposed for diagnosing coronavirus from a person’s chest CT scans or chest X-ray imaging [21]. The suggested model is a layered ensemble before heterogeneous ML models. Some of the transfer learning techniques are VGG-19, ResNet, DenseNet, and fully connected networks are the pre-trained DL models. By adjusting the extra fully connected tiers in each of them before the model, prospective candidates for the classification algorithm are obtained. After an extensive search, three outstanding models were chosen to create approximate weighting heterogeneity stacked ensembles. Early diagnosis of coronavirus-positive cases prevents the disease from spreading further in the population and allows for earlier treatment of the general public [22]. CT imaging and chest X-ray images have recently shown distinctive characteristics that indicate the severity of COVID-19 in the lungs. The advancement of AI technology, specifically DL-based healthcare, is crucial in handling vast amounts of data to achieve accurate and quick outcomes in diagnostic imaging. It can also aid in the more efficient and precise diagnosis of pathogens while assisting in remote locations. Various two-dimensional convolution techniques have been proposed for analyzing chest X-ray images to identify COVID-19 with a precision of 99% and a validation accuracy of 98%, with a loss of around 0.15%.

Rapid identification of coronavirus [23] was a challenging and time-consuming task that computer-aided diagnosis (CAD) techniques have effectively addressed, enabling quick decision-making and meaningful tasks for surveillance, treatment, and medication. The use of chest X-ray (CXR) technique has become a cost-effective and practical alternative to CT scanning and real-time polymerase chain reaction (RT-PCR) testing, which is the most widely used detection method for COVID-19. Although several CAD approaches have been developed for detecting COVID-19, various issues have hindered their effectiveness. This research develops a deep CNN (DCNN) model using ResNet32 with the convolutional block attention module to detect COVID-19 and pneumonia.

VGG-16 surpasses all these models on the dataset with an accuracy of 85.33% [24]. Clinicians employing RT-PCR tests can diagnose COVID-19, but it has a high rate of false positive (FP) and false negative (FN) findings, and it takes a long time. One way to increase the true positive (TP) ratio is to run many experiments simultaneously. CT scans and X-ray images can be used to identify COVID-19-related pneumonia earlier. Upwards of 95% efficiency can be reached using current DL methods. Various CNN (CovNet)-based DL models are applied to diagnose pneumonia, including ResNet 152 v2, InceptionResNet v2, Xception, Inception v3, ResNet 50, NASNetLarge, DenseNet 201, and VGG 16 [25].

A DL framework can be created to assist in evaluating CT tests to offer diagnostics that can help save time in managing the disease. In this study, a DL algorithm was adapted to identify COVID-19 using extracted features from chest X-rays and CT images. Many transfer learning models were initially used and compared, and a VGG-19 model was subsequently tweaked to produce the best results for disease diagnosis. The diagnosis accuracy of all models was evaluated using a sample of 1000 images. The VGG-19 model has higher precision, sensitivity, and specificity, with 99.4% precision, 97.4% sensitivity, and 99.4% specificity in the timely identification of coronavirus. ML and image analysis performed well [26]. Researchers present an approach to examining the possibility of deep transfer learning in developing a classification to identify coronavirus-positive patients utilizing CT and CXR imaging in this research. The preprocessing data method expands the training datasets to lessen overfitting and improve the model’s generalization capacity. To use a preprocessing data method, we evaluated pre-trained DL models: ResNet50, InceptionV3, VGGNet-19, and Xception. DL is better at finding coronavirus cases, according to the research. The VGGNet-19 model surpasses the other three models presented using the CT image dataset when each modality is considered separately, according to the findings of the tests [27]. It is true that Deep Neural Networks (DNNs) have made significant progress in computer vision tasks, including medical imaging, and DenseNet has been evaluated for categorizing COVID-19 chest X-ray images. The COVID-19 pandemic has highlighted the importance of medical imaging in the diagnosis and monitoring of diseases, and machine learning algorithms such as DNNs can help automate and streamline the analysis process. DenseNet is a popular convolutional neural network architecture that has shown promising results in various image classification tasks. A public database of 6432 chest X-ray images divided into three classes was also used. The three DenseNet modeling variants, DenseNet121, DenseNet169, and DenseNet201, were trained via transfer learning and fine tuning. After analyzing the results, it was found that DenseNet201 had the best accuracy rate of 0.9367 and the most negligible validation loss of 0.1653 for COVID-19 classifications in chest X-ray images [27,28].

Patients must wait an extended period of time to receive the results of blood tests to detect COVID-19. This method uses a DL technique to detect COVID-19 quickly. DL and neural algorithms logistic regression is a statistical tool for predicting the outcome of radiology data, such as CT scans and X-ray images, that are fed into these algorithms. Using this approach, positive instances of COVID-19 will be recognized more quickly [29]. This research focuses on building a DL approach for detecting and diagnosing COVID-19 utilizing chest CT scans. A public database was employed for this, which included 349 CT scans of 216 patients with COVID-19 clinical abnormalities and CT scans of 397 healthy people. Accuracy, precision, recall, Matthews coefficients correlation (MCC), and F-measure criteria were used to evaluate diagnostic accuracy. A 10-fold cross-validation procedure was used to assess the reliability of the approach. CNN has an accuracy rate of 92.63%, a precision of 92.95%, a recall of 93.18%, an MCC of 85.2%, and an F1 measure of 93.06%. Based on the results achieved with the approach taken within the focus of this research, the mentioned approach could be used as a supplement or replacement for traditional therapeutic tests in coronavirus outbreaks [29,30]. For the identification and classification of COVID-19, a transfer learning method with precise adjustment was used in this study. VGG16, DenseNet-121, ResNet-50, and MobileNet were employed as before models. DL models were developed using a dataset of 7232 (COVID-19 vs. healthy) chest X-ray images. A local dataset of 450 chest X-ray images from Pakistani patients was gathered or used for tests and predictions. Many essential characteristics, such as recall, specificity, F1-score, accuracy, loss graphs, and confusing matrix, were generated to verify the accuracy of the algorithms. VGG16, ResNet-50, DenseNet-121, and MobileNet attained an accuracy of 83.27%, 92.48%, 96.49%, and 96.48%, respectively [31].

This research aimed to show how DL, ML, and image processing can help with the fast and accurate identification of COVID-19 from two of the most frequently used neuroimaging techniques, chest X-rays and CT scans. CNN models based on the Alexnet model were proposed for CAD of coronavirus from CT and X-ray images. The effectiveness of this technique was tested on the obtained dataset. COVID-19 CT and X-ray scans were categorized, and CT scans were used to track the progression of the disease [32]. COVID-19 has been successfully identified by utilizing CT scans and X-rays to assess lung images. However, it takes a team of radiologists and a lot of time to review each report, which is time-consuming. As a result, this work proposes an automatic COVID-19 recognition and categorization algorithm based on DNNs. The provided method performs pre-processing, extraction of features, and classifications. The input image is processed using median filtering at an earlier stage. The VGGNet-19 model, built on a CNN, is then used as a feature representation. Finally, to identify and classify the presence of COVID-19, an ANN is used as a classification algorithm [33]. Researchers worldwide are investigating new ML approaches, such as DL, to help medical experts diagnose COVID-19 disease using medical information such as X-ray films and CT scans. ML techniques for demonstrating the first COVID-19 identification from chest X-rays would have been effective in controlling this epidemic, because the facility for chest X-rays is available even in smaller cities and is significantly less expensive. As a result, researchers suggest a CNN for identifying the presence and absence of COVID-19. The CNN model is compared with the other models based on ancient and transfer learning. Compared to ML algorithms, the suggested CNN is more efficient (KNN, SVM, DT, etc.) [34,35]. COVID-19 may be identified using DL techniques, a crucial task for treatment possibilities identified based on data these days. The development of AI, ML, DL, and diagnostic imaging approaches offers excellent performance, particularly in the areas of detection, classification, and segmentation. The advances in imaging technologies that have allowed clinicians to monitor the body and accurately diagnose and assess patients without surgery. It also highlights the use of several imaging technologies, including CT with a DL model built on a CNN, for detecting COVID-19 [36,37,38,39,40,41,42,43].

## 3. Methods

This section focuses on the need for the proposed system model in detail.

### 3.1. Need for Proposed Model

The proposed model is required to fulfill the following requirements: to obtain better measures, and to automate optimum model architecture and training parameters in COVID-19 and pneumonia diagnosis; to apply the transfer learning approaches to solve content gaps and shorten training duration; to perform multi-class classification for X-ray imaging tasks by applying an upgraded VGG16 deep transfer learning; to minimize the complexity of the layered design to perform the multi-class categorization of various illnesses.

### 3.2. System Model

This section explores the proposed model as sketched in Figure 1. The spatial domain procedure is applied to the image pixels and altered pixel by pixel in this filtering. It is a filtering mask that shifts from one pixel to another by performing several operations. It will remove noise from the image by smoothing it.

The spatial domain filtering is divided into linear and non-linear filters. The mean and Wiener filters are the most often used linear filtering techniques, whereas the median filter is a non-linear filtering approach. The mean value in the mean filter is derived from the computation of neighbors and center pixel values in the N × N size. It is calculated after calculating the center pixel value as follows:(1)$$Y\left(M,N\right)=Mean\;\left\{X\left[i,j\right],\;\left[i,j\right]\epsilon \;w\right\}$$
where w are the pixel positions in the neighborhood.

The Wiener filter is applied for filtering the consistent pixel values that define the constant power additive noise. This filtering technique is used for adaptive filtering of the image pixel-wise. Two variations are computed by computing the pixels in the neighborhood with the following mean and standard deviation:(2)$$\mu =\frac{1}{N\times M}\sum_{\left(n_{1},n_{2}\in \eta \right)}a\left(n_{1},n_{2}\right)$$
(3)$$\sigma^{2}=\frac{1}{N\times M}\sum_{\left(n_{1},n_{2}\in \eta \right)}a^{2}\left(n_{1},n_{2}\right)-\mu^{2}\;$$
where η represents the N×M of the current pixel; with this estimation, the pixel-wise Wiener filter was applied over the denoised image, which is computed as follows:(4)$$b\left(n_{1},\;n_{2}\right)=\mu +\frac{\alpha^{2}-v^{2}}{\alpha^{2}}\left(a\left(n_{1},n_{2}\right)-\mu \right)$$
where $v^{2}$ represents the noise variance.

The median filter works by the pixels in the window sorted in ascending order. The median value of the N×M image is changed based on the central pixel values using
(5)$$Y\left(M,N\right)=Median\;\left\{X\left[i,j\right],\;\left[i,j\right]\epsilon \;w\right\}$$
where w is the pixels of the neighborhood.

Image enhancement uses quality enhancement techniques to improve the image contents based on essential features, including intensity, edges, and corners. Histogram equalization redistributes the pixel values of an image to increase contrast, making it easier to distinguish between different image features. This technique can help to reduce the number of pixels that are saturated or redundant in certain areas of the image, but its primary goal is to enhance the overall visual quality of the image. This process increases the contrast level of the image and the computation speed of content-based medical image retrieval. Histogram equalization is a technique that improves the image’s visual quality by adjusting the noisy and blurred pixel values. The histogram represents the discrete function or the intensity distribution with the graphical format applied to dark images. The process of histogram equalization for contrast enhancement is defined as follows:Read the input image.Compute the histogram, probability of each pixel, and probability density function of the input image.Equalize the histogram.

The histogram represents the image’s intensities using bars representing pixel frequency. For a given input image, the histogram is computed as follows:(6)$$H\left(r_{k}\right)=n_{k}$$
where $r_{k}$ is the $k^{th}$ gray level value ranging from 0 to L-1, and $n_{k}$ is the number of pixels in the image which have the gray level in $r_{k}$. Then, determine the probability of the gray level using
(7)$$p\left(r_{k}\right)=\frac{n_{k}}{M\times N}$$
where $n_{k}\;$is the number of pixels, M×N defines the image size with M rows and N columns. The pdf of the image is computed using
(8)$$p_{k}=\overset{k}{\underset{i=0}{\sum }}n_{k}\times \frac{1}{M\times N}$$

Then, the histogram equalization is applied to enhance the quality of the image by correcting its intensity values for the discrete case using
(9)$$S_{k}=\;\left(L-1\right)\times p_{k}\left(r\right)$$
where $p_{k}(r$) represents the pdf.

### 3.3. Fast Attention-Based ResNet

Attention-based ResNet is used to perform feature extraction and classification. The architecture of the proposed model is depicted in Figure 2. The learning rate of the proposed model is set to 0.0001, and the number of epochs performed for classification is 10. The extracted set of features has many irrelevant and redundant features that need to be removed because it reduces the accuracy of the process. Then, the feature selection is performed to remove the irrelevant and redundant characteristics [7]. The attention module is responsible for the extraction of features by learning the weights of the features corresponding to the scaling attacks. The output of the attention layer is computed using
(10)$$Atn_{i}\left(Z\right)=Q_{i}\left(Z\right).\;F_{i}\left(Z\right)+F_{i}\left(Z\right)$$
where $Q_{i}\left(Z\right)$ denotes the attention weight and$\;F_{i}\left(Z\right)$ denote the features. In the attention layer, the relationship between the features is computed to achieve more relevant information from the features, which can be computed as,
(11)$$R\left(F_{1};F_{2}\right)=\overset{n}{\underset{F_{1}}{\int }}\overset{n}{\underset{F_{2}}{\int }}p\left(F_{1},F_{2}\right)\log\frac{p\left(F_{1},F_{2}\right)}{p\left(F_{1}\right)p\left(F_{2}\right)}dF_{1}dF_{2}$$
where $p\left(F_{1},F_{2}\right)$ denote the function of probability between the features $F_{1}$ and $F_{2}$ and $p\left(F_{1}\right)$, $p\left(F_{2}\right)$ denote the individual function of marginal density, respectively. The inception layer learns specific features deeply by reducing the size of the initial patch, which affects the original information required for the classification process. Based on the selected features, classification is carried out. This process ascertains whether a given unknown packet is standard or malicious. The softmax layer performs this classification, in which the cross entropy is computed to determine the output loss, which can be calculated as
(12)$$Loss\;\left(x\right)=-\sum_{k}\log\left(st{\left(x\right)}_{k}\right)q_{i}$$

Formulate the softmax $st\left(x\right)$ of a vector using
(13)$$st{\left(x\right)}_{k}=\frac{e^{x_{k}}}{\sum_{m}e^{x_{m}}}$$

The classification is performed by increasing the score corresponding to the input images.

### 3.4. Enhanced VGG-16 Architecture for X-ray Images

Figure 3,the VGG-16 architecture is implemented to extract and classify the features using DL [8]. Initially, the features of the segmented region are extracted. The proposed VGG-16 has three layers: a convolutions layer, a fully connected layer, and a softmax layer. VGG 16 extracts low-level features with small masks and has fewer layers than VGG 19. The max pooling and average pooling layers extract the features from the image. These layers extract the features from the segmented region, as shown in Table 1. The convolution layer concatenates the pooling results using the sigmoid function$\left(\delta \right)$ with the measure of
(14)$$C\left(F\right)=\delta \left(f^{7\times 7}\left[F_{A};F_{M}\right]\right)$$
where $F_{A}\in r^{1\times h\times w}\;$and $F_{M}\in r^{1\times h\times w}\;$represent expected and max pooling measures.

The fully connected layer receives the extracted results using dense, flattened, and dropout layers to bring the final classification as normal or abnormal using
(15)$$C\left(p=q/s\right)=\frac{e^{b}\;}{\sum_{j}e^{b_{j}}}$$

Algorithm 1 presents the segmentation process of dividing an image into multiple segments or regions based on some criteria, such as color, texture, or shape. VGG16 is a convolutional neural network that has been widely used for image classification tasks. However, it can also be used for feature extraction and segmentation tasks.
**Algorithm 1: VGG 16** **Input:** Segmented Region (S.R.) **Output:** Normal (N) or abnormal (AN) { **Begin** $\mathrm{Initialize}\;\left\{fc,\;fs,\;ft\right\}$ $fc\leftarrow \left\{fc1,fc2,..fcn\right\}$ $fs\leftarrow \left\{fs1,fs2,..fsn\right\}$ $ft\leftarrow \left\{ft1,ft2,..ftn\right\}$ **Initialize** the feature extraction data. **for **
i←o to n
**do** **Extract**
 fc from S.R. **Extract**
 fs from S.R. **Extract**
 ft  from S.R. $F\leftarrow \left\{fc,fs,ft\right\}$ **Extract** the features by average pooling layer
$F_{A}$ **Extract** the features by maxpool layer
$F_{M}$ **Concatenate** the features **Classifying** the images using softmax layer $\mathrm{Class}\leftarrow \left\{\;N,\;AN\right\}$ **end for** } **return** class **end** }

## 4. Results and Discussion

This section analyzes the experimental results of the proposed method.

### 4.1. Dataset

Five different databases are applied to evaluate the performance of the proposed method. Two databases provide chest X-ray imaging, while the remainder include chest CT scans [11,12]. Every database was divided into three parts: test, validation, and training set. The test set must have between at least 200 and 400 images to evaluate the model’s flexibility well. The size of the test set determines the size of the verification set; the more significant the test set, the more prominent the testing set will be, and conversely. The other images were used to develop a training set. To perform the presented approach, the test and the validation test sets are organized, consisting of the same partitions of the positive and negative image samples. The hyper-parameters are tuned based on the training and test sets of the coronavirus images.

The descriptions of the datasets are given as follows:

The CT image dataset contains approximately 349 CT images from 216 patients with coronavirus and 397 patients without coronavirus. The images were collected from hospitals that treat both coronavirus and non-coronavirus patients, but only the positive and negative cases for coronavirus were included in the dataset [13,14]. Some of the samples of coronavirus images in both positive and negative classes are given in Figure 4. The features of the dataset are as follows:

Types of Images: CT ImagesSize of Dataset: 746 CT ScansPositive Case images in Total: 349Negative Case images in Total: 397Validation Size Set: 118 ScansTraining Size Set: 425 ScansTest Size Set: 203 Scans

The *COVID-19 images collection dataset* consists of images gathered from the public community, such as physicians and hospitals. The features of this dataset are as follows:

Types of Imaging: Chest X-raysSize of Dataset: 579 ImagesTraining Size Set: 309Validation Size Set: 70Testing Size Set: 200COVID-19 Positive Cases: 342COVID-19 Negative Cases: 237

The *CT set COVID-19 dataset* consists of original CT scan imaging for 377 patients. A total of 15,589 CT scans are used for CT scan images of 95 coronavirus patients and 282 regular patients. The features of this dataset are as follows:

Types of Images: CT Scan ImagesSize of Dataset: 12,058 ScansPositive Images: 2282Negative Images: 9776Training Set: 11, 400Validation Set: 258Testing Set: 400

The COVID-19 radiography collection includes 1200 images of positive COVID-19, 1341 images of healthy patients, and 1345 images of patients with fever infection. This dataset has the following characteristics:

Figure 5 CT scans, or computed tomography scans, use X-rays to create detailed images of the inside of the body. CT scans can be helpful in diagnosing respiratory illnesses such as SARS-CoV-2, the virus that causes COVID-19. CT scans can show the presence of lung abnormalities such as ground-glass opacities and consolidation, which can be indicative of viral pneumonia.

Images Type: X-raySum of Images: 2541Negative Images: 2686Positive Images: 1200Training Size Set: 3086Testing Size Set: 400Validation Size Set: 400

*SARS-CoV-2 CT scan images* consists of 1252 CT scan images that are positive, whereas the remaining 1230 images are classified as non-infected viruses, with the following features:

Images Type: CT Scan ImagesSize of Dataset: 2482 ImagesNegative Images: 1230Positive Images: 1252Training Size Set: 1800Testing Size Set: 400Validation Set Size: 400

To mitigate the model overfitting issues of the training case, the training size is increased from 1275 images collected utilizing data augmentation based on random rotation, horizontal flipping, and color jittering.

### 4.2. Data Preprocessing and Augmentation Techniques

Using data augmentation methods, some enormous volumes of datasets must be tested using transfer learning or DL techniques [14,15]. Hence, this research model considers data augmentation methods with the datasets.

Resize or Crop by Random: This step represents the cropping of the input image to the unsystematic size of the image and the aspect ratio.Rotation by Random: To perform this step, the sample is rotated by selecting an angle at random.Horizontal Flip by Random: This step represents the flip action for the given input image randomly in a horizontal manner.Color Jittering: This step randomly represents the modifications of the input image’s contrast, saturation, and brightness.Training Settings: The overall work is implemented through a Python framework, in which the variables are fixed for all experiments. The simulation parameters are shown in Table 2.

### 4.3. Comparison Analysis

Classification accuracy is defined as the accuracy in performing the classification of both COVID-19 and viral pneumonia and computed using
(16)$$Accuracy=\frac{T_{r}P+T_{r}N}{T_{r}P+T_{r}N+F_{l}P+F_{l}N}\times 100\%$$

It is important to find the optimal number of iterations that maximizes the model’s accuracy without overfitting. This can be done by monitoring the validation accuracy during training, which measures how well the model performs on a separate set of data that is not used for training. The validation accuracy typically starts to plateau after a certain number of iterations, indicating that the model is no longer improving significantly and further training may lead to overfitting in Figure 6.

The greater the classification accuracy, the greater the efficiency of the approach. Figure 6 and Figure 7 depict the relationship between the model’s classification accuracy with others based on the iterations and the sample counts. The classification accuracy of the proposed ensemble model is high due to the extensive extraction of features and classification.

The feature set is generated from many features using a pre-trained model to eliminate redundant features [16,17]. The selection of transfer learning in the model is based on the recent study and literature analysis, contributing to increased classification accuracy. The existing approaches considered both necessary and redundant features for classification, which degraded the classification accuracy.

The classification accuracy of the proposed method with others based on users and iterations count is presented in Table 3. The classification and accuracy of the proposed method are 95% and 96%, respectively, whereas the existing approaches possess only about 62% to 72% of accuracy. The proposed method accurately classifies COVID-19 and non-COVID-19 images.

The precision is the measure of relevancy in the classified images computed using
(17)$$\mathrm{Precision}=\frac{T_{r}P}{T_{r}P+F_{l}P}\times 100\%$$

The comparison of the precision of the presented method and the existing approaches to iteration count and the number of samples is illustrated in Figure 8 and Figure 9. The figures show that the precision of the proposed model is high due to the implementation of spatial domain filtering, thereby eliminating the noise. Increasing the iteration count of a machine learning model can lead to improvements in recall, but this is not always the case Figure 10.

Further, the classification of the face and non-face regions was performed by utilizing the feature extraction in which the optimum approach performed the selection of features. The lack of consideration of pixel-based features and the removal of the noise level in the images of the existing approaches resulted in reduced precision [17,18]. The superiority of the proposed pre-trained approach is illustrated in Table 4, which presents the numerical comparison of the presented method and existing approaches in terms of the iteration count. The presented method’s precision is 95% to 96%, whereas the existing approaches possess precision of about 61% to 71%. The proposed method produces better precision than other methods in detecting coronavirus.

The recall is called sensitivity, the measure of correctness between the classified images, calculated using
(18)$$Recall=\frac{TP}{TP+FN}$$

Figure 11 and Figure 12 sketches the comparison of recall measure with other methods over the fixed iterations in all cases of test data. The recall of the presented method is high due to the implementation of effective pre-processing techniques to eliminate the noise and correct the illumination.

Further, consideration of various features, such as low and high-level features, improved the recall results of the proposed model. The lack of noise elimination and consideration of integrated features restricted the recall of the existing approaches. The numerical analysis of recall with other methods is shown in Table 5. The presented method has a recall of 94% to 96%, whereas the existing approaches possess only about 62% to 72%. This leads to the conclusion that the presented method is more efficient than the existing approaches in performing coronavirus detection in healthy and viral infection cases [20].

The F-measure is the combination of precision and recall calculated by computing the harmonic mean and measured using
(19)$$F-measure=\frac{2\times TP}{\left(2\times TP+FP+FN\right)}$$

The comparison of the F-measure with other methods based on the iteration count is illustrated in Figure 13. The increased F-measure of the proposed method is due to the increase in precision and recall value.

The detection and classification are carried out by pre-processing the input frames and extracting extensive features from the pre-processed images. The grouping of pixels is performed to determine the differentiation between the non-similar features. The classification of coronavirus and pneumonia is determined with an increased F-measure over the reduced F-measure of the other methods.

The efficiency of the presented method is proved by the numerical analysis presented in Table 6, which compares the F-measure with the other methods. The F-measure of the presented method is 95% to 97%, whereas the existing methods range from 63% to 72%. The model that was developed has been shown to be efficient for pre-training purposes in the detection and classification of masked faces.

The Receiver Operating Curve (ROC) depicts the relationship between the specificity and sensitivity in all possible cut-off values. The ROC curve is a graphical representation to visualize the efficiency of an approach. The better performance of an approach is depicted as the curve nearer to the left top corner of the ROC curve. The ROC curve of the proposed model with others is also analyzed for all the image sets of evaluation. The proposed model yields a better ROC curve than the other models in classification and detection [36,37]. The existing approaches possess reduced ROC curve values due to the inefficiency caused by image noise and the lack of feature integration for classification. The computational time is the time to compute specific tasks to obtain the desired result. Figure 13 compares the computational time of the proposed and other models based on the iteration factor. The computational time involved with the existing approaches is high due to the increased time consumption in training the model.

As the number of samples increases, the computation time required to process those samples also increases. This is because more samples typically mean more data to process and analyze, which can be computationally intensive. For example, if you’re training a machine learning model on a dataset of images, the more images you have, the longer it will take to train the model Figure 14.

The benefits of the proposed model are as follows: the pre-trained model possesses fewer layers than the existing models and thereby possesses low complexity. This characteristic of the proposed model requires a short computational time to detect and recognize the coronavirus.

Figure 15, this method provides minimal computation time without compromising the accuracy in detecting and recognizing the coronavirus [38,39]. The time complexity of the layered architecture runs in a linear, that is, O(n) time. The proposed model is computationally efficient and highly accurate for the multi-class categorization of distinct illness types [40,41,42,43].

Table 7 shows the comparison of the computation time with other models. From the table, it is clear that the computation time of our approach is shallow, at about 0.4 s. In contrast, the existing approaches possess an increased computation time of up to 0.9 s for both preprocessed and non-preprocessed.

### 4.4. Limitations and Analysis

The limitations of the model are as follows: quality of outcomes based on the noise levels in the images affect the performance measures; integrating the relevant features are also an important requirement for the classification in the proposed model. The future research directions of the proposed model are as follows: enhancement using recent soft computing components and better recommender systems will be developed based on the features of datasets [19,20,44,45,46,47,48]; large datasets will be considered for validation to further improve performance measures using soft computing within the minimal computing time [25,26,27,49,50,51,52,53,54,55,56,57,58,59,60,61,62,63].

## 5. Conclusions

COVID-19 and pneumonia are usually encountered due to lung symptoms that can be identified through genes and studies. Early detection of coronavirus and effective management of its progression can be facilitated through the use of imaging tests. In identifying coronavirus disease, chest X-rays and CT are helpful imaging modalities. The wide availability of huge, annotated images based on transfer learning methods—VGG-16, ResNet, and DenseNet—has led to considerable progress in earlier transfer learning models for medical image classification. This ensemble learning technique is an accurate representation because it recovers a hierarchy of localized visual elements given the input. However, irregularities in the annotation of COVID-19 and pneumonia instances produced on X-ray imaging remain a much more problematic component of working on them. This research classified COVID-19 and pneumonia images in a vast chest X-ray and CT database using an ensemble learning system based on developments in the extraction features technique. The suggested methodology provided fast and thorough COVID-19 and pneumonia classification results and the ability to deal with data inconsistencies and a limited amount of class images. The classification and accuracy of the proposed method are about 95% and 96%, respectively, and the precision is found to be 95% to 96%. The recall is 94% to 96%, with an F-measure from 95% to 97%. The proposed method is more efficient than the existing approaches in performing coronavirus and pneumonia detection in healthy and viral infection cases. The devised model is adequate for pre-training in detection and classification. The time complexity of the layered architecture runs in a linear, that is, O(n) time, and takes less than 0.5 s. The proposed model is computationally efficient and highly accurate for the multi-class categorization of distinct illness types.

## Figures and Tables

**Figure 1 bioengineering-10-00363-f001:**
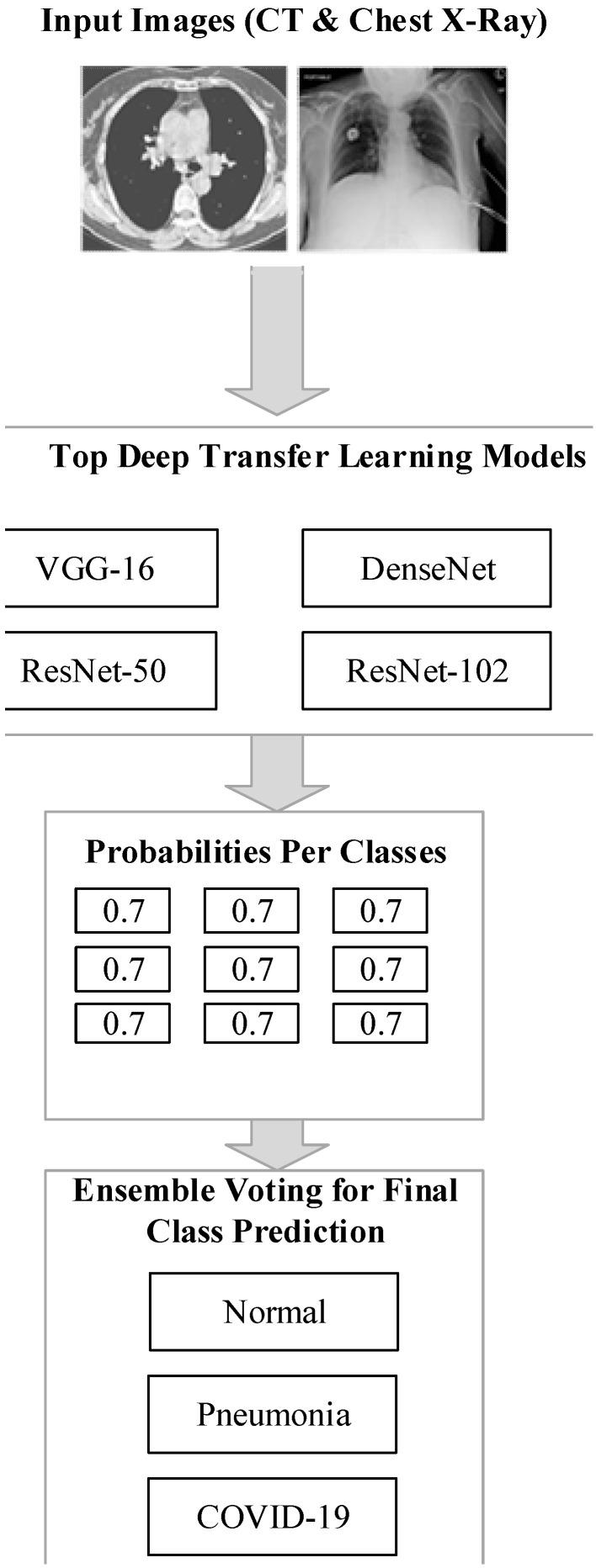
The architecture of the proposed model.

**Figure 2 bioengineering-10-00363-f002:**
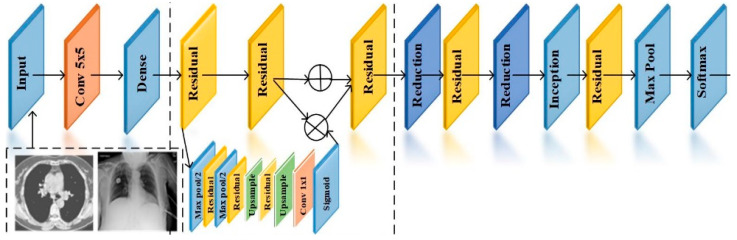
Attention-based ResNet.

**Figure 3 bioengineering-10-00363-f003:**
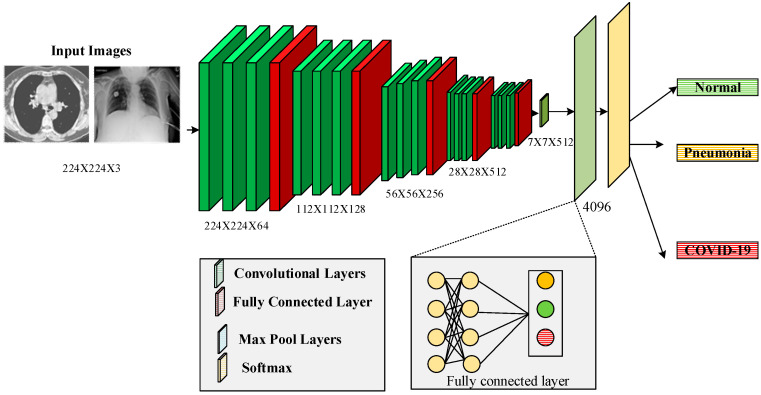
The architecture of VGG 16.

**Figure 4 bioengineering-10-00363-f004:**
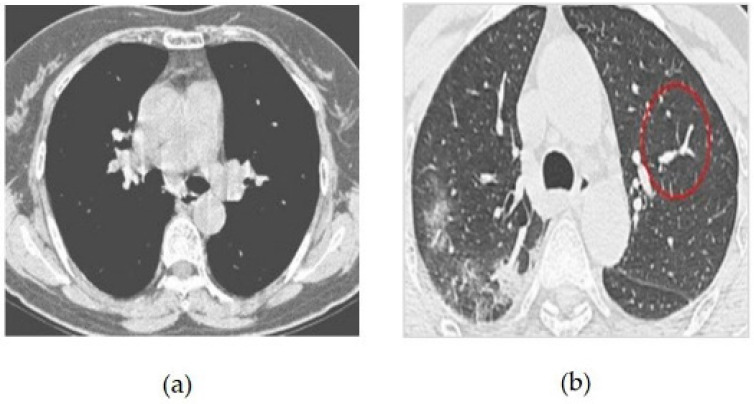
(**a**) COVID-19 Negative (COVID-19 images collection dataset) (**b**) COVID-19 Positive (COVID-19 images collection dataset).

**Figure 5 bioengineering-10-00363-f005:**
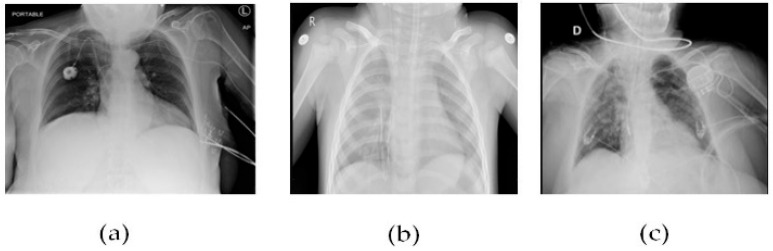
(**a**) Patient healthy image, (**b**) Patient with pneumonia, and (**c**) Patient with +Ve COVID-19 images COVID-19 negative.

**Figure 6 bioengineering-10-00363-f006:**
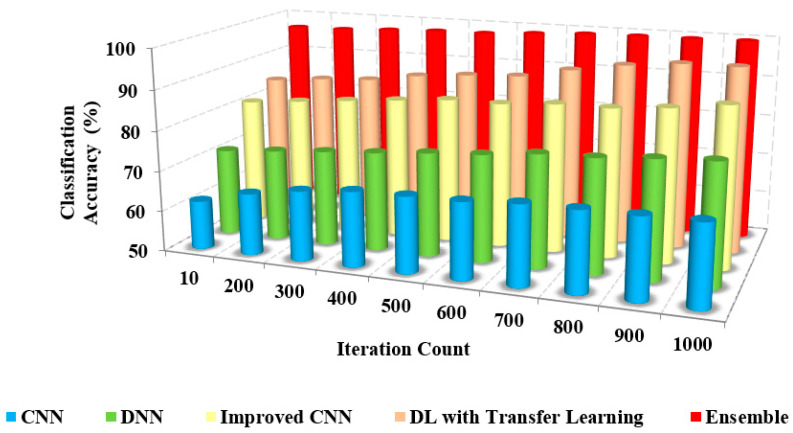
Iteration count vs. classification accuracy.

**Figure 7 bioengineering-10-00363-f007:**
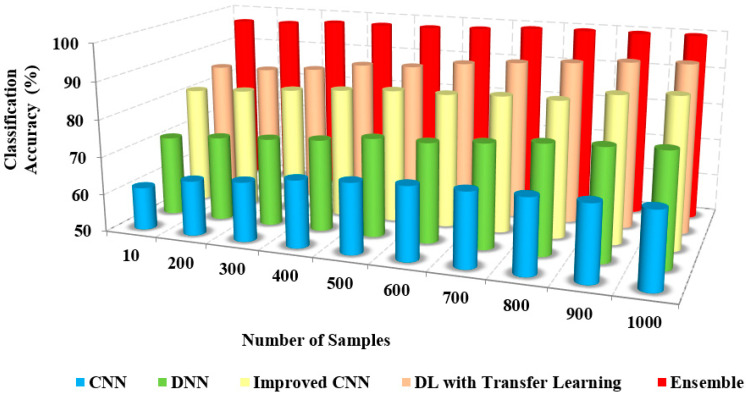
Number of samples vs. classification accuracy.

**Figure 8 bioengineering-10-00363-f008:**
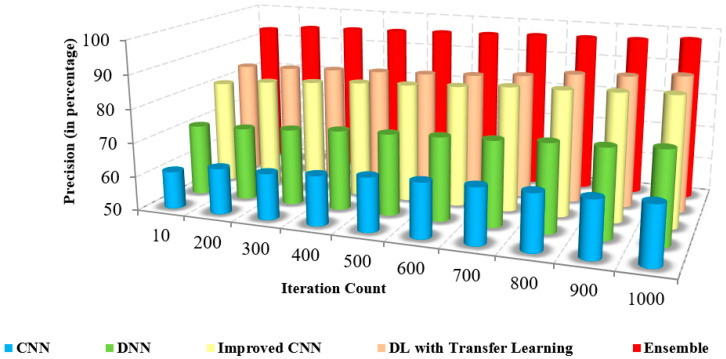
Iteration count vs. precision.

**Figure 9 bioengineering-10-00363-f009:**
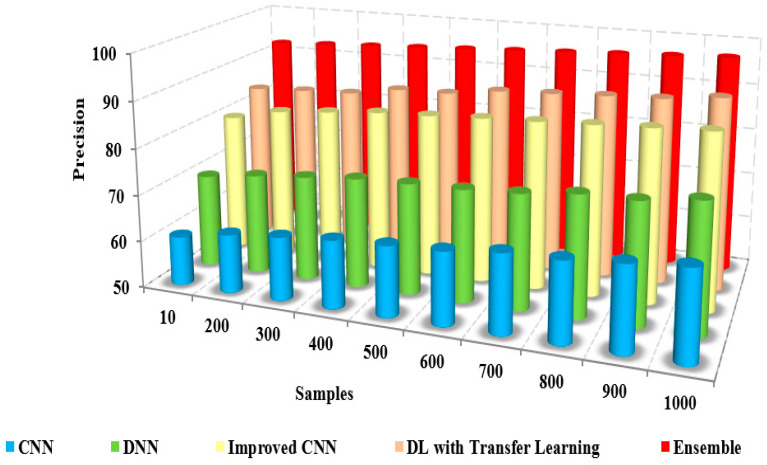
Samples vs. precision.

**Figure 10 bioengineering-10-00363-f010:**
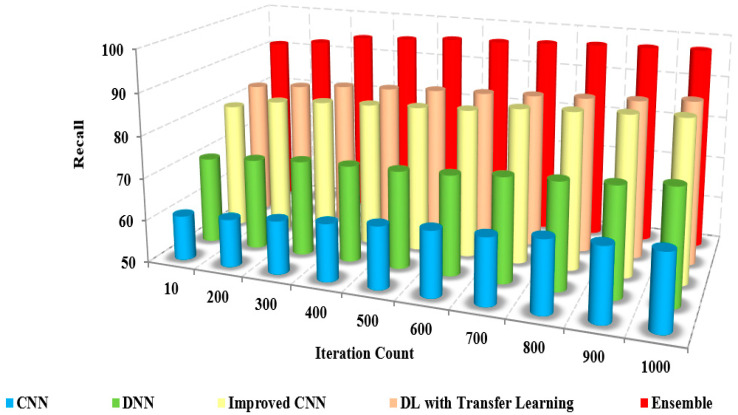
Iteration count vs. recall.

**Figure 11 bioengineering-10-00363-f011:**
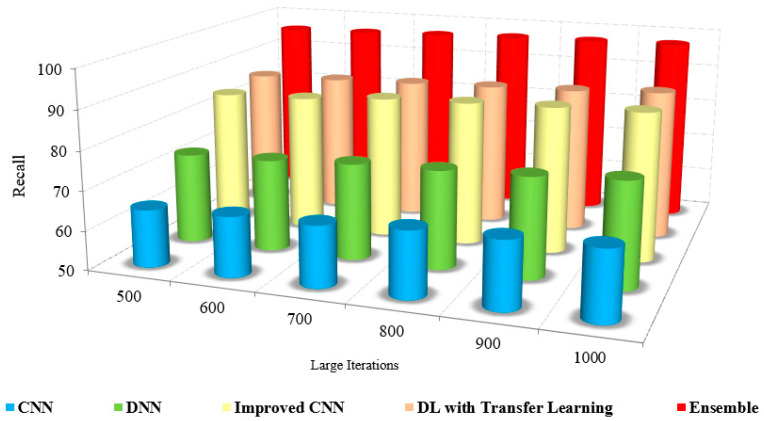
Large Iteration count vs. recall.

**Figure 12 bioengineering-10-00363-f012:**
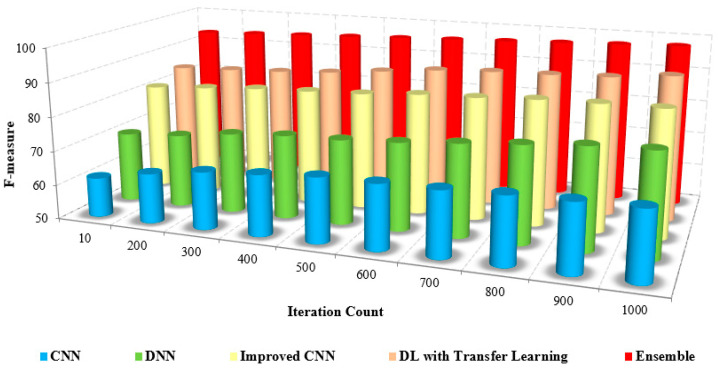
Iteration count vs. F-score.

**Figure 13 bioengineering-10-00363-f013:**
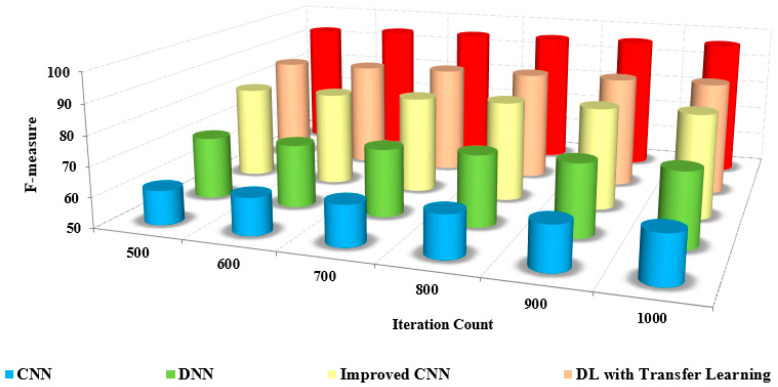
Large iteration count vs. F-score.

**Figure 14 bioengineering-10-00363-f014:**
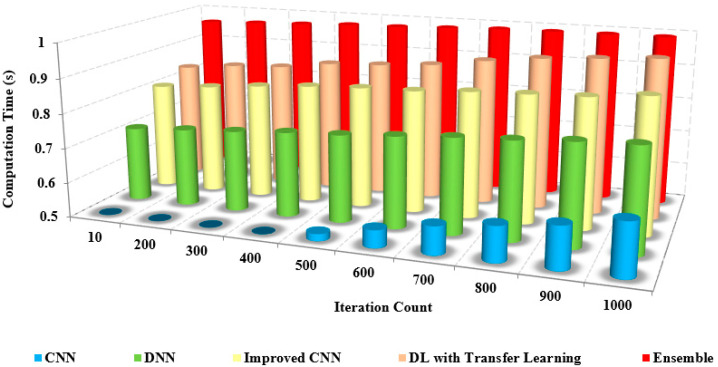
Iteration count vs. computation time.

**Figure 15 bioengineering-10-00363-f015:**
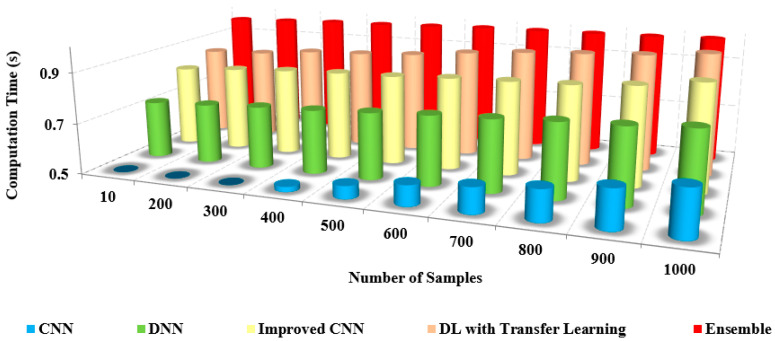
Number of samples vs. computation time.

**Table 1 bioengineering-10-00363-t001:** Features and Aesthetics of the model.

Color	Standard deviation, intensity, kurtosis, skewness
Texture	Homogeneity, entropy, correlation
Shape	Artery—radius, length, angle, area, eccentricity

**Table 2 bioengineering-10-00363-t002:** Simulation parameters.

Parameters	Values
DL Model	Python (PyTorch library)
Number of Epochs	1 to 100
Learning Rate	0.003
Optimization Technique	ADAM
Loss	Cross Entropy
Size of Batch	16
Resized Cropping Scale in Random	0.5 to 1.0
Rotation Angle Range	(−5, +5)
Flipping Possibility Rate	0.5
Resized Size of Cropping in Random	224

**Table 3 bioengineering-10-00363-t003:** Classification accuracy.

Methods	Classification Accuracy (%)	
	Iteration Count	No of Samples
CNN	61 ± 0.05%	62 ± 0.04%
DNN	71 ± 0.02%	72.6 ± 0.03%
Improved CNN (LeNet-5)	84 ± 0.03%	85.5 ± 0.04%
DL with Transfer Learning	92 ± 0.02%	92.5 ± 0.1%
Ensemble	95 ± 0.2%	96.2 ± 0.5%

**Table 4 bioengineering-10-00363-t004:** Precision.

Methods	Precision (%)	
	Iteration Count	Number of Samples
CNN	60 ± 0.05%	62 ± 0.04%
DNN	71 ± 0.02%	72.6 ± 0.03%
Improved CNN (LeNet-5)	84 ± 0.03%	85.5 ± 0.04%
DL with Transfer Learning	92 ± 0.02%	92.5 ± 0.1%
Ensemble	95 ± 0.2%	96.2 ± 0.5%

**Table 5 bioengineering-10-00363-t005:** Recall.

Methods	Recall (%)	
	Iteration Count	No. of Samples
CNN	59.6 ± 0.05%	61.5 ± 0.04%
DNN	69.8 ± 0.02%	71.5 ± 0.03%
Improved CNN (LeNet-5)	84 ± 0.03%	83.5 ± 0.04%
DL with Transfer Learning	91.5 ± 0.02%	92 ± 0.1%
Ensemble	94.5 ± 0.2%	95.9 ± 0.5%

**Table 6 bioengineering-10-00363-t006:** F-measure.

Methods	F-Measure (%)	
	Iteration Count	No. of Samples
CNN	60.2 ± 0.05%	62.3 ± 0.04%
DNN	71.2 ± 0.02%	72 ± 0.03%
Improved CNN (LeNet-5)	84.5 ± 0.03%	84.2 ± 0.04%
DL with Transfer Learning	92.2 ± 0.02%	92.5 ± 0.1%
Ensemble	95.2 ± 0.2%	96.7 ± 0.5%

**Table 7 bioengineering-10-00363-t007:** Computation time (s).

Methods	Computation Time	
	Masked Face	Unmasked Face
CNN	0.9	0.96
DNN	0.84	0.92
Improved CNN (LeNet-5)	0.78	0.84
DL with Transfer Learning	0.65	0.72
Ensemble	0.4	0.62

## Data Availability

Not applicable.

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
