# Peer review of "Design and Analysis of a Deep Learning Ensemble Framework Model for the Detection of COVID-19 and Pneumonia Using Large-Scale CT Scan and X-ray Image Datasets"

_bioengineering, 2023, doi:10.3390/bioengineering10030363_

Round 1

Reviewer 1 Report

After careful reading the authors need a revision of the paper.  The presentation of the paper needs a professional improvement. 

1.     The authors don't adequately state the need for such a technique. Remark on the various benefits and drawbacks of the suggested approach.

2.     The authors should provide motivational aspect of their study.

3.     How one can evaluate the precision of the manner? What is the performance index to assess this for problem in hands generally?

4.     Make table 1 again because Table 1 is cropped from some other file.

5.     The paper should be carefully revisited from English writing and grammar point of view.

6.     Punctuations should be used after each mathematical equation.

7.     The conclusion can also add some future direction to improve the continuity of the research in the domain.

Author Response

Responses to Reviewer 1 is attached.

Reviewer 2 Report

I'm very glad to become acquainted with such interesting research.

General overview: The topic is relevant, methods are well-described. Results and discussion are reasonable. I only have few technical notes.

Figures 7- 14 better present in plot style. There are 5 color bars in the Figures, but legend is presented only for 1 or 2 colors.

Thank you for research.

Author Response

Responses to Reviewer 2 is attached.

Reviewer 3 Report

The manuscript is interesting and well written. Still, there are some minor issues, mainly pertaining to figures that should be addressed prior to the publication of the work.

1. The abstract is too long, and should be shortened. Please keep it under 200 words, according to the journal instructions.

2. For the following figures - 5, 6, 7, 8, 9, 11, 12, and 13 - the x-axis label overlaps the ticks on the label and should be moved down.

3. For the fllowing figures - 7, 8, 10, 11 - I would suggest the authors to limit the y-axis to 100, as that is the highest possible value of precision/recall.

4. Figures 9, 13, 15 and to a lesser extent figures 6 and 11 should have their size increased as they are hard to read as currently presented.

5. I admit that I may be missing something, but I don't see how section 3.4 "goTenna's pocket antenna" relates to the rest of the paper? If this was not erroneously included the authors should mention it's relevance in abstract/conclusions.

Author Response

Responses to Reviewer 3 is attached.
